# Evolution of Allogeneic Stem Cell Transplantation: Main Focus on AML

**DOI:** 10.3390/cells14080572

**Published:** 2025-04-10

**Authors:** Yoshitaka Inoue, Joseph Cioccio, Shin Mineishi, Kentaro Minagawa

**Affiliations:** 1Blood and Marrow Transplant Program, Division of Hematology and Oncology, Penn State Cancer Institute, Hershey, PA 17033, USA; jcioccio@pennstatehealth.psu.edu (J.C.); smineishi@pennstatehealth.psu.edu (S.M.); kminagawa@pennstatehealth.psu.edu (K.M.); 2Cancer Institute, Penn State College of Medicine, Hershey, PA 17033, USA

**Keywords:** acute myeloid leukemia (AML), allogeneic stem cell transplantation (Allo-HSCT), measurable residual disease (MRD), post-transplant cyclophosphamide (PTCy), TP53 mutations, pretransplant and post-transplant strategies, GVHD prophylaxis, maintenance therapy

## Abstract

In recent years, treatments in the field of hematologic malignancies have undergone significant evolution; allogeneic hematopoietic stem cell transplantation (allo-HSCT) has shifted from an “ultimate” therapy to becoming a component of a comprehensive therapeutic strategy for acute myeloid leukemia (AML). Advances in risk stratification (including molecular profiling and measurable residual disease assessment), conditioning regimens, and graft-versus-host disease (GVHD) prophylaxis—such as post-transplant cyclophosphamide—have improved outcomes and expanded donor selection and transplant eligibility. We should not only focus on the transplantation procedure but also consider various therapeutic components, including chemotherapy, targeted therapy (possibly including chimeric antigen receptor T-cell therapy), and post-transplant maintenance therapy, which need to be orchestrated within the broader context of leukemia treatment. In this review, we summarized key developments in allo-HSCT for AML and aim to “decipher” each component of transplantation.

## 1. Introduction

In recent years, many new treatments have become available in the field of hematologic malignancies. Together with these revolutionary changes, allogeneic stem cell transplantation (allo-HSCT) has also been evolving in its role within the therapeutic scheme. Allo-HSCT is no longer considered the “ultimate” therapeutic modality. Instead, it is now regarded as a component of a comprehensive therapeutic strategy (Figure 1). Various therapeutic components, such as chemotherapy, targeted therapy (possibly including chimeric antigen receptor T-cell; CAR-T), and post-transplant maintenance therapy, need to be orchestrated within the broader context of leukemia treatment.

Additionally, the latest advances, such as measurable residual disease (MRD) determination and molecular genetics, should be incorporated into each step of the decision-making process. In this review, we aimed to “decipher” each component of transplantation, with a primary focus on acute myeloid leukemia (AML).

## 2. Transplant Indications

### 2.1. Indication for Transplantation in AML

Fit patients with relapsed AML are candidates for transplantation. But who requires transplantation in the first complete remission (CR)? Historically, allo-HSCT was recommended for patients with a poor prognosis when treated with chemotherapy alone on the basis of their cytogenetics. While transplant outcomes can vary with each center [1], individual institutions may have their own criteria for transplantation. Nonetheless, some consensus exists regarding indications for transplantation. Today, prognostic groups incorporate both cytogenetics and molecular features, most notably as defined by the 2022 recommendations from the European Leukemia Net (ELN2022) [2], with survival outcomes delineated for each risk group. Generally, patients with poor cytogenetics should be considered for transplantation [2]. According to the ELN2022, cytogenetic abnormalities associated with poor prognosis include

t(6;9)(p23.3;q34.1)/DEK::NUP214;t(v;11q23.3)/KMT2A-rearranged (excluding partial tandem duplication);t(9;22)(q34.1;q11.2)/BCR::ABL1;t(8;16)(p11.2;p13.3)/KAT6A::CREBBP;inv(3)(q21.3q26.2) or t(3;3)(q21.3;q26.2)/GATA2, MECOM (EVI1);t(3q26.2;v)/MECOM (EVI1)-rearranged;−5 or del(5q), −7, −17 or abn(17p);complex karyotype and monosomal karyotype.

Patients with adverse-risk mutations should also be considered for transplantation, including those with:

TP53 biallelic mutations [3,4], TP53 high-variant allele frequency (VAF) [5], or TP53 with complex cytogenetics [6].

Mutations in myelodysplasia-related genes, such as ASXL1, RUNX1, BCOR, EZH2, SF3B1, SRSF2, STAG2, U2AF1, or ZRSR2 [2]

High allelic ratio FMS-like tyrosine kinase 3 (FLT3)-internal tandem duplication (ITD) with wild-type nucleophosmin 1 (NPM1) was previously classified as a poor prognosis group [7]. However, with the development of FLT3 inhibitors, FLT3-ITD is now classified as an intermediate prognosis group, regardless of the allelic ratio or the presence of NPM1 mutation status [2].

Patients in the intermediate-risk groups were once considered for transplantation in the first CR [8]. However, this became controversial after an EBMT randomized study was published [9]. The study randomized 143 intermediate-risk AML patients in first CR to either allo-HSCT or conventional consolidation chemotherapy. The overall survival (OS) was not superior in the transplantation arm compared to the chemotherapy arm (74% vs. 84% at two years, *p* = 0.22). However, disease-free survival (DFS) was superior in the transplantation arm (69% vs. 40% at two years, *p* = 0.001). The reason for the lack of OS superiority, despite significantly better DFS, was that all the relapsed patients in the chemotherapy arm underwent transplantation as salvage therapy. This study demonstrated that first CR transplantation is not mandatory if salvage transplantation is well-prepared following relapse. Therefore, selecting patients for transplantation in the first CR should involve not only the intermediate-risk classification but also individual factors such as age, performance status, comorbidities, donor availability, and patient preference.

Finally, MRD positivity after treatment is now considered an indication for transplantation. This topic will be discussed in detail in the “MRD” section.

### 2.2. Indication for Transplantation in Myelodysplastic Syndrome (MDS)

The historical article by Cutler et al. defined transplant indications for patients with MDS based on the International Prognostic Scoring System (IPSS). According to their criteria, patients with intermediate-2 or higher risk should undergo transplantation, whereas those with intermediate-1 or lower risk may experience shortened survival from the procedure [10]. Since that publication, clinicians have used an intermediate-2 risk score or approximately a 3-year expected survival as the transplant cutoff, recommending transplantation if the expected survival is shorter.

For now, the IPSS-M would be the most advanced risk stratification tool for MDS, incorporating molecular risk factors [11]. Following the same principle, for each MDS-related disease, a projected survival of less than 3 years based on disease-specific risk stratification guides transplant decisions.

### 2.3. Indication for Transplantation in Myelofibrosis

The Dynamic International Prognostic Scoring System (DIPSS) is the standard for determining transplant eligibility, as it is used for defining Medicare coverage [12]. However, DIPSS-plus or the Mutation-Enhanced International Prognostic Scoring System for patients aged ≤ 70 years (MIPSS70), which also incorporate molecular risk factors, are alternative scoring systems [13,14]. Notably, MIPSS70 provides refined risk stratification by including gene mutation profiles, and its intermediate- and high-risk categories show prognostic similarities to DIPSS intermediate-1 and intermediate-2, respectively. As with IPSS-M, the integration of molecular data enhances prognostic accuracy and may guide individualized transplant decisions more effectively.

### 2.4. Indication for Transplantation in Chronic Myelomonocytic Leukemia (CMML)

The CMML-specific Prognostic Scoring System incorporating molecular data (CPSS-mol) is generally employed to guide transplant decisions [15]. CPSS-mol Int-2 estimates their median OS as 30–36 months, which could be controversial for the indication of transplantation. EBMT Practice Harmonization and Guidelines Committee recommends one or more additional risk factors in those with Int-2, including extramedullary disease, leukocytosis, iron overload, splenomegaly, and adverse cytogenetics and/or gene mutations, should proceed to allo-HSCT; otherwise, they can proceed to a watch and wait strategy [16]. Also, the keynote message in this paper is that one should proceed directly to allo-HSCT without disease modification therapy unless truly needed.

## 3. Transplant Survival Improvement

In general, transplantation is recommended if it offers better survival outcomes than conventional therapies. Over the past 20 years, transplant survival rates have improved significantly [17,18]. Consequently, expanded transplant indications now reflect these improved outcomes. However, since transplantation can cause long-term complications, such as graft-versus-host disease (GVHD), it is essential to evaluate both survival outcomes and post-transplant quality of life (QOL) when determining eligibility.

GVHD-free, relapse-free survival (GRFS) is a composite key metric that measures survival without relapses or severe GVHD following transplantation [19,20]. This metric helps assess not only extended survival but also the patient’s ability to maintain a good QOL. In recent years, GRFS has become a common primary endpoint in clinical trials, replacing traditional measures such as OS or progression-free survival (PFS). Additionally, some clinical decision analyses incorporate QOL metrics to guide transplant eligibility assessments [21].

Many transplant centers now use post-transplant cyclophosphamide (PTCy) for GVHD prophylaxis. PTCy has been shown to reduce the incidence and severity of GVHD and outperform conventional GVHD prophylaxis in improving GRFS. Consequently, PTCy has contributed significantly to enhancing patients’ quality of life post-transplant [22,23].

## 4. Challenges in Transplant Practice

### 4.1. MRD Positive Diseases

In AML, MRD positivity after induction therapy is considered an indication for transplantation. MRD positivity can be detected through various methods, including flow cytometry, cytogenetics, fluorescence in situ hybridization, or molecular mutation analysis (Table 1). Regardless of the method, MRD positivity correlates with a 60% or higher relapse rate [24,25]. Post-transplant relapse rates for non-remission AML cases typically range from 60% to 80% [26,27,28]. Emerging evidence suggests that MRD-positive AML in remission may have similar relapse rates to non-remission AML. This is because the inflection point for relapse risk may be far below the 5% blast cutoff traditionally used to define remission—potentially around 0.1% [29,30,31].

#### 4.1.1. Defining MRD Sensitivity

When discussing MRD-positive diseases, it is crucial to consider the sensitivity of MRD detection methods. Multi-color flow cytometry typically detects MRD down to 0.1% [32]. Molecular methods, such as qPCR, can reach sensitivities of 10^−5^, while next-generation sequencing at research levels can achieve 10^−6^ sensitivity [2]. However, commercial gene panels may only detect down to 1–3%, highlighting significant differences in detection capabilities [33]. MRD positivity as a transplant indication generally corresponds to 0.1% detection levels. However, if a highly sensitive assay, such as ClonoSEQ (currently used mainly for lymphoid malignancies), detects MRD at 10^−6^, the decision to proceed with transplantation should be carefully individualized.

#### 4.1.2. Approaches for MRD-Positive Disease

Myeloablative conditioning (MAC) generally offers better outcomes than reduced-intensity conditioning (RIC) for MRD-positive AML, Ref. [34] although some studies report conflicting results [35]. Notably, studies showing 60% relapse rates for MRD-positive patients included many cases treated with MAC regimens, suggesting that MAC alone may be insufficient to address MRD. One promising strategy involves incorporating preconditioning regimens. For example, using clofarabine preconditioning has been shown to reduce tumor burden by 5 logs at the time of transplantation [36]. This significant reduction in disease burden could make MRD-positive cases more manageable, potentially achieving near-elimination of the disease. Further details on this approach will be discussed in a subsequent section.

### 4.2. TP53 Mutations

TP53 is a tumor suppressor gene located on chromosome 17p13.1. Mutations in TP53 occur in approximately 10–20% of AML patients and are associated with poor prognosis [37]. Although several retrospective studies have demonstrated the efficacy of transplantation in AML patients with TP53 mutations [38,39], outcomes remain unsatisfactory, and no prospective studies have been conducted to evaluate transplant effectiveness.

Not all TP53 mutations carry the same risk. Particularly poor prognostic factors include biallelic TP53 mutations [3,4], TP53 with high VAF [5], and TP53 mutations with complex cytogenetics [6]. Without transplantation, most patients with these high-risk mutations succumb to the disease within a short time. Even with transplantation, outcomes remain suboptimal [40,41]. Recent studies suggest that certain subsets of patients with TP53 mutations may benefit from transplantation [42]. For instance, according to a report from the EBMT, transplant outcomes in TP53-mutated AML patients without complex karyotype or 17p deletion were comparable to those in patients with preserved TP53 function [43]. Similarly, another study showed improved survival with transplantation in patients with a VAF ≤ 40%, but not in those with higher VAF [5]. On the other hand, there is currently no conclusive evidence supporting the benefit of TP53 mutation clearance prior to transplantation in patients with TP53-mutated AML, and further investigation is needed to clarify this issue [39,44].

In transplant for TP53-mutated AML/MDS, additional factors associated with poor prognosis include impaired performance status and the presence of comorbidities. Ciurea et al. identified three clinical variables—Hematopoietic cell transplantation (HCT)-specific comorbidity index > 4, Karnofsky Performance Status (KPS) ≤ 80%, and advanced disease beyond first or second CR—as independent predictors of inferior post-transplant survival outcomes [45]. Furthermore, patients with TP53 mutations have also been reported to have a higher risk of pre-transplant infections, such as bacterial pneumonia and invasive fungal infections, and a higher rate of infection-related mortality compared to those without such mutations, despite a comparable duration of neutropenia [46]. Current evidence indicates that transplant outcomes in patients with TP53 mutations are not significantly influenced by donor type or stem cell source [43,45]. Importantly, MAC regimens have not demonstrated a survival advantage over RIC in this population [47,48,49].

Recent clinical trials have explored novel therapeutic approaches targeting TP53 mutations. Eprenetapopt (APR-246), a TP53-specific inhibitor combined with hypomethylating agents (HMAs), has shown promise. In a phase Ib/II trial, the CR rate was approximately 50% in MDS patients and 30–40% in AML patients, with improved survival for those who proceeded with transplantation after eprenetapopt plus azacitidine therapy [50,51]. Post-transplant maintenance, with this combination, also demonstrated encouraging results [52]. Additionally, magrolimab, an anti-CD47 agent, in combination with azacitidine, has shown efficacy in patients with TP53-mutated AML/MDS [53,54]. However, neither eprenetapopt nor magrolimab has demonstrated efficacy in phase III trials, highlighting the ongoing challenges in treating TP53-mutated AML/MDS.

TP53-mutated AML/MDS remains one of the most challenging subtypes to treat. A single therapeutic modality is unlikely to be sufficient; instead, a comprehensive, multi-modal approach may be necessary. Further clinical trials exploring innovative combination strategies are essential to improve outcomes in this high-risk patient group.

### 4.3. Non-Remission AML

Patients with non-remission AML present significant challenges for allo-HSCT, with relapse rates comparable to those in remission with MRD positivity; thus, they may belong to the same category. The strategy successful in non-remission AML could provide a solution for those in remission with MRD. After two failed induction attempts, the treatment for AML patients becomes more challenging [55,56]. Patients with non-remission AML need a transplant to survive, but many transplant centers in the United States do not even offer transplants due to historically poor results [26,57,58].

#### 4.3.1. CloBu4 Trial and Outcomes

We conducted a clinical trial using clofarabine (Clo) and busulfan (Bu) for four days (CloBu4) in relapsed/refractory AML patients undergoing allo-HSCT. The day +30 remission rate was 94% for all non-remission AML and 100% for first-time transplant patients, with a one-year overall survival of 48% and 2-year DFS of around 20% [59]. A multicenter study with the same regimen confirmed these results, particularly benefiting patients with primary induction failure, who achieved a 2-year DFS of approximately 40% [60].

#### 4.3.2. Pre-Conditioning Approach

Preconditioning, where induction chemotherapy is administered soon before transplant conditioning begins, has shown success, particularly in Europe [61,62]. The idea of preconditioning is as simple as starting the transplant conditioning before malignant cells recover after induction chemotherapy. A recent prospective study comparing standard re-induction followed by transplant versus preconditioning demonstrated comparable outcomes, validating this approach [63]. Some of the preconditioning studies utilized clofarabine [64,65].

#### 4.3.3. Clofarabine Preconditioning

We have used CloBu4 with only HLA-matched donors. When we had non-remission AML patients with no HLA-matched donors available, we used clofarabine as a “pre-conditioning” and haploidentical transplant was initiated three days after the clofarabine treatment (Figure 2) [66]. Initially, we used fludarabine (Flu) and Bu for 3 days (FluBu3) as the conditioning regimen, but it was toxic to even relatively young patients, so we changed the conditioning regimen to Flu and 2 days of Bu (FluBu2). Alternatively, we employed Flu plus 800 cGy of total body irradiation (TBI) with PTCy for several patients, and modified cyclophosphamide (Cy) plus TBI in other cases (Cy 20 mg/kg, TBI 800 cGy, then PTCy). PTCy is being used not only for haploidentical transplant but universally as well, as GVHD was moderate to severe if PTCy was not used, possibly due to further tissue damage and cytokine release due to preconditioning. If the patient’s age is over 60 or the patient is unfit, we used Clo preconditioning with FluCyTBI 200 cGy followed by PTCy (Johns Hopkins RIC regimen). This regimen was well-tolerated by elderly patients but the relapse rate was higher compared to more intensive transplant regimens. Currently, we use Clo preconditioning followed by Flu, melphalan (Mel), and PTCy—with Mel 140 mg/m^2^ for younger patients and Mel 100 mg/m^2^ for older patients—with better results (Manuscript in preparation).

### 4.4. Post-Transplant Maintenance Strategies

Studies on the efficacy of post-transplant maintenance with HMAs alone have produced mixed results [67,68,69,70]. However, emerging evidence suggests that maintenance therapy may provide benefits in specific subtypes, such as AML with FLT3-ITD, particularly when using FLT3 inhibitors [71,72,73,74,75] (Table 2). Notably, a European study demonstrated significant improvement in DFS with post-transplant sorafenib, a FLT3 inhibitor [72]. Similarly, results of the BMT-CTN 1506 trial showed that post-transplant maintenance with gilteritinib improved relapse-free survival in patients who were MRD-positive at the time of transplant, but not in those who were MRD-negative [74]. With rapid advances in molecular therapeutics, effective post-transplant maintenance therapies may soon benefit a broader range of AML patients. It should be noted that quizartinib has been approved in Europe by the European Medicines Agency as a maintenance therapy after transplant in FLT3-ITD-positive AML, but it has not been approved for this indication by the U.S. Food and Drug Administration (FDA).

The current concept emphasizes that allo-HSCT, per se, is not sufficient to cure the majority of leukemia patients; rather, it may serve as a central platform in a comprehensive treatment scheme that effectively integrates pre- and post-transplant therapies (Figure 1). Importantly, planning for early initiation of post-transplant maintenance therapy is crucial and may influence the choice of conditioning regimen. Although MAC is generally associated with a lower risk of relapse compared to RIC, this advantage may be offset by higher non-relapse mortality, resulting in no significant difference in OS. Therefore, in clinical practice, avoiding intensive MAC regimens could be considered in selected cases to enable timely initiation of maintenance therapy. Additionally, the use of PTCy may facilitate early maintenance by reducing the risk of acute GVHD [76,77].

**Table 2 cells-14-00572-t002:** Maintenance therapy after allogeneic stem cell transplant in acute myeloid leukemia.

Agents	Drug Name	Author (Study ID)	Disease	N, Study Design	Results	*p*-Value
Flt3 inhibitors	Sorafenib	Burchert et al. [72] (SORMAIN /DRKS00000591)	FLT3-ITD^+^ AML	Sorafenib (N = 43) Placebo (N = 40), Phase II	2-year RFS 85% in Sorafenib 53.3% in Placebo	<0.01
		Xuan et al. [78] (NCT02474290)	FLT3-ITD^+^ AML	Sorafenib (N = 100) Placebo (*n* = 102), Phase III	1-year RR 7% in Sorafenib 24.5% in Placebo	<0.01
	Gilteritinib	Levis et al. [74] (BMT−CTN 1506 /NCT02997202)	FLT3-ITD^+^ AML	Gilteritinib (N = 178) Placebo (N = 178), Phase III	2-year RFS 77.2% in Gilteritinib 69.9% in Placebo	0.052
	Midostaurin	Maziarz et al. [73] (RADIUS /NCT01883362)	FLT3-ITD^+^ AML	Midostaurin (N = 16) SOC (N = 14), Phase II	18-month RFS 89% in Midostaurin 76% in SOC	0.27
	Quizartinib	Sandmaier et al. [79] (2689-CL-0011/(NCT01468467)	FLT3-ITD^+^ AML	N = 13, Phase I	RR 7.7%	-
		Erba et al. [75] (QuANTUM-First/NCT02668653)	FLT3-ITD^+^ AML	Quizartinib (N = 268) Placebo (N = 271), Phase III	Median OS 31.9 months in Quizartinib 15.1 months in Placebo	0.03
	Crenolanib	Oran et al. [80] (NCT02400255)	FLT3-ITD/TKD^+^ AML	N = 30, Phase II	5-year OS 69% 5-year RFS 69.7%	-
IDH1/2 inhibitors	Ivosidenib	Fathi et al. [81] (NCT03564821)	IDH1(R132)-mutant AML	N = 18, Phase I	1-year RR 19%	-
	Enasidenib	Fathi et al. [82] (NCT03515512)	IDH2-mutant AML, MDS or CMML	N = 23, Phase I	2-year OS 74% 2-year PFS 69% 2-year RR 16%	-
		Salhotra et al. [83] (NCT03728335)	AML with IDH2 mutation	N = 15, Phase I	2-year OS 100% 2-year LFS 100%	-
Bcl-2 inhibitor	Venetoclax	Kent et al. [84]	AML	N = 49, Phase II	1-year OS 70% 1-year PFS 67% 1-year RR 20%	-
		Wei et al. [85] (ChiCTR1900025374)	AML/MDS	N = 20, AML (N = 17) MDS (N = 3), Phase II	2-year RFS 84.7% (median follow-up 598 days, treated with low-dose decitabine)	-
		Gracia et al. [86] (NCT03613532)	AML/MDS/MPN	N = 27, AML (N = 10), MDS (N = 16), MPN (N = 1), Phase I	2-year OS 67% 2-year PFS 59% 2-year RR 41%	-
Hypomethylating agents	5-azacytidine (AZA)	Oran et al. [67] (NCT00887068)	AML/MDS	AZA (N = 93) Control (N = 94), Phase III	Median RFS 2.07 years in AZA 1.28 years in control	0.14
		Keruakous et al. [68]	AML	AZA (N = 31) Control (N = 18), Phase II	RR 25.8% in AZA 66.7% in Control	<0.01
	Oral formulation of AZA (CC-486)	De Lima et al. [87] (NCT01835587)	AML/MDS	7 days per cycle (*n* = 7), 14 days per cycle (*n* = 23), Phase I	1-year RFS 54% in 7-day and 72% in 14-day CC-486 dosing	-
	Decitabine (DAC)	Pusic et al. [88] (NCT00986804)	AML/MDS	N = 22, Phase I	2-year OS 56% 2-year DFS 48% 2-year RR 28%	-
		Gao et al. [69] (ChiCTR-IIR-16008182)	AML	G-Dec (*n* = 100) Non-G-Dec (*n* = 102), Phase II	2-year RR 15% in the G-Dec 38.3% in the non–G-Dec	<0.01
		Ma et al. [70]	AML	DAC = 21 Control = 63, Retrospective study	3-year RFS 94.1% in DAC 55% in control	<0.01

Abbreviations: AML—Acute myeloid leukemia, BMT-CTN—The Blood and Marrow Transplant Clinical Trials Network, CMML—Chronic monocytic myelogenous leukemia, DAC—Decitabine, DFS—Disease-Free Survival, FLT3—Fms-like tyrosine kinase 3, G-Dec—Recombinant human granulocyte colony stimulation factor and decitabine treatment, IDH—Isocitrate dehydrogenase, ITD—Internal tandem duplication, LFS—Leukemia-Free Survival, MDS—Myelodysplastic syndrome, NCT—National Cancer Trial, OS—Overall Survival, PFS—Progression-Free Survival, RFS—Relapse-Free Survival, RR—Relapse Rate, SOC—Standard of Care, TKD—Tyrosine kinase domain mutation.

### 4.5. Role of Consolidation Chemotherapy in the Era of Reduced Intensity Conditioning

For some patients who require allo-HSCT, such as elderly patients, unfit patients, or those with certain disease subtypes, it is advisable to proceed to transplant without prior intensive chemotherapy, which may increase transplant-related comorbidities. During the era of full-intensity conditioning, it was relatively clear that consolidation therapy provided no advantage if the patient was proceeding directly with the transplant [89]. In the current era, with a focus on RIC, the benefit of consolidation chemotherapy remains unclear and, at best, controversial [90,91,92,93]. Consolidation therapy may help prevent disease recurrence while awaiting a donor, especially since there is a possibility that a suitable donor may not be found promptly. For example, venetoclax, either alone or in combination with azacitidine, may serve as a bridge to transplant for patients with aggressive diseases or when a suitable donor is not immediately available [94,95]. However, with the routine use of PTCy, haploidentical donors, and mismatched unrelated donors, donor searches have become significantly faster, and only in rare cases is a donor unavailable for a patient. Therefore, a relatively mild consolidation regimen may be more appropriate to maintain remission until transplant without increasing toxicity.

### 4.6. How PTCy Changed the Transplant Practice

PTCy was initially developed for haploidentical bone marrow transplant at Johns Hopkins [96]. The principle of this strategy is that donor T-cells stimulated by host antigens enter the cell cycle, resulting in alloreactive T-cell proliferation by day 3. On day 3 and 4, Cy eliminates those proliferating donor T-cells; only dormant T-cells survive. Since stem cells are resistant to high-dose Cy due to their higher level of aldehyde dehydrogenase, which inactivates Cy, they are able to engraft normally [97,98]. This enzyme also drives regulatory T-cell resistance to Cy leading to prevention of GVHD [99,100,101,102]. PTCy is considered the most potent GVHD prophylaxis, enabling safe haploidentical transplantation. Notably, this prophylaxis is effective beyond haploidentical donors. The HOVON96 prospective European study evaluated PTCy for matched related and unrelated transplants, showing comparable survival between PTCy and conventional GVHD prophylaxis groups. Additionally, the PTCy group demonstrated superior GRFS [22]. Then, the BMT-CTN 1703 study confirmed these findings with a larger cohort of patients [23]. We performed a retrospective analysis at our institution and found that PTCy had significantly better survival outcomes than conventional GVHD prophylaxis among all alternative donors. The conventional GVHD prophylaxis group included 237 patients who received allo-HSCT from a matched unrelated donor (*n* = 214) or mismatched unrelated donor (*n* = 23). The PTCy group consisted of 64 patients, with unrelated matched (*n* = 16), mismatched (*n* = 20), and haploidentical donors (*n* = 28). The OS was better in the PTCy group (79.7% vs. 54.8% at 2-year OS, *p* = 0.01, Figure 3A). The relapse rate tended to be lower in the PTCy group than in the conventional GVHD prophylaxis group (17.8% vs. 32.4% at 2-year relapse rate, *p* = 0.08). PTCy also reduced the number of patients dying from GVHD, effectively eliminating GVHD-related mortality. In our study, GVHD-related mortality in the PTCy group was significantly lower than in the conventional GVHD prophylaxis group (0% vs. 7%, *p* = 0.03, Figure 3B) [103]. Originally, PTCy was used only for haploidentical bone marrow transplants. However, accumulating evidence indicates that any donor–recipient combination can be a candidate for the PTCy regimen. Since relapse/refractory AML and increased tissue damage may elevate the risk of GVHD, we incorporated PTCy into the preconditioning regimen with Clo.

A challenge with PTCy is its intensity. Older patients or those with comorbidities, particularly cardiac conditions, may not tolerate it well. Combined with transplant conditioning, it imposes a significant additional burden. We used a single dose of 50 mg/kg/day PTCy for matched unrelated donor transplants in a previous clinical trial [104] but the incidence of GVHD was comparable to that with conventional GVHD prophylaxis [96]. Recently, PTCy with 25 mg/kg/day × 2 day regimen is being tested [105] and may be promising in terms of producing similar GVHD-preventing efficacy with PTCy at 50 mg/kg/day × 2. Compared to a previous study of ours with a single-dose regimen [104], the two-day administration may be more effective even at the same total dose. Furthermore, PTCy 25 mg/kg/day × 2 may help decrease engraftment failures, which we will discuss below, compared to PTCy 50 mg/kg/day × 2.

Adverse events other than GVHD, such as engraftment failure, delayed engraftment, infections, and multi-organ failure, contribute to early post-transplant mortality. The focus of transplantation may be shifting from GVHD prevention to the prevention and treatment of engraftment failure and infections [106,107,108]. While PTCy ablates alloreactive donor T-cells, memory T-cells, which preferentially expand post-transplant [109], may increase the risk of graft failure, especially in patients with prior alloimmunization [98].

Our group identified large spleen size, particularly in patients with myeloproliferative disorders including CMML, as a significant risk factor for engraftment failure, especially in PTCy transplants [110]. We now obtain non-contrast CT of the abdomen to measure spleen size before PTCy transplants and may consider alternative GVHD prophylaxis in case of severe splenomegaly (greater than 20 cm in craniocaudal length).

Table 3 compares methods for preventing engraftment failure. Alemtuzumab and anti-thymocyte globulin suppress both donor and recipient T-cell function, with unclear effects on engraftment. In contrast, low-dose (200 cGy) TBI reduces engraftment failure by targeting recipient cells only. We compared donor chimerism kinetics and survival with and without TBI in all PTCy cases at our institution (Figure 4) [111]. Although TBI does not affect overall survival (1-year overall survival in the low-dose TBI group 80.4% vs. the non-TBI group 76.8%, *p* = 0.6), it improved donor chimerism in T cells, especially in unrelated donor cases (Figure 4). For patients with large spleens, we added spleen irradiation in addition to TBI.

### 4.7. Advances in GVHD Prophylaxis Beyond PTCy

GVHD prophylaxis has advanced beyond PTCy, and various preventive strategies are now being utilized [112]. Anti-thymocyte globulin (ATG) has long been used, particularly in Europe, due to its potent T-cell-depleting effects that prevent GVHD. Recently, studies comparing ATG and PTCy as GVHD prophylaxis in AML patients have been reported [113,114]. These reports suggest that in transplants performed in CR1 from HLA-matched related donors or unrelated donors, transplant outcomes—including the incidence of GVHD—are comparable between the ATG and PTCy groups.

Abatacept is a cytotoxic T lymphocyte-associated 4 immunoglobulin G1 fusion protein that prevents T-cell activation by inhibiting co-stimulatory signaling between CD28 and CD80/86, thereby reducing the risk of GVHD [115]. In a phase II trial evaluating GVHD prophylaxis using calcineurin inhibitor/methotrexate plus abatacept in unrelated donor transplantation for patients with hematologic malignancies, including AML, the addition of abatacept significantly reduced the incidence of acute GVHD compared to the regimen without it [116]. Based on the results of this trial, abatacept has been approved by the FDA for the prevention of acute GVHD. Furthermore, a recently published study using the Center for International Blood and Marrow Transplant Research data reported that the outcomes of the calcineurin inhibitor/methotrexate plus abatacept regimen were comparable to those of PTCy in unrelated donor transplants [117]. On the other hand, the current four-dose schedule of abatacept (10 mg/kg on Day −1, +5, +14, and +28) has raised concerns about an increased risk of chronic GVHD. The results of ongoing trials evaluating extended dosing of abatacept are eagerly awaited (NCT04380740).

Another promising strategy for GVHD prophylaxis involves Orca-T and Orca-Q, which are investigational cell-engineered therapies. Orca-T contains hematopoietic stem cells and highly purified donor-derived regulatory T-cells that suppress alloreactive immune reactions [118,119,120]. It is manufactured in a centralized GMP-compliant laboratory and distributed to multiple centers across the United States. A Phase Ib study has reported the outcomes of transplant from HLA-matched donors using Orca-T, combined with single-agent Tac, in 37 AML patients in CR. The NRM and OS at 12 months after transplant were 0% and 100%, respectively, demonstrating highly promising results [121]. Moreover, Orca-Q is a cell therapy product composed of selected T-cell subsets and hematopoietic stem cells, and has shown promising results in HLA-matched transplant even without additional GVHD prophylaxis [122].

### 4.8. Transplant for Elderly AML Patients Aged 70 and Above

With the advent of RIC regimens, safer and more effective GVHD prophylaxis (e.g., PTCy), improvements in transplant outcomes from alternative donors, and advances in supportive care as discussed later, the indications for transplant have expanded to include patients up to approximately 75 years of age [123,124]. A large retrospective study including elderly AML patients aged 60 to 77 years in CR1 reported improved leukemia-free survival in those who underwent transplantation compared with patients who received chemotherapy alone [125]. Although transplant provides meaningful benefits, its application in AML patients aged ≥ 70 years remains challenging. According to a report from the EBMT comparing transplant outcomes between AML patients aged ≥ 70 and those aged 50–69, the 2-year NRM was significantly higher in patients aged ≥ 70 (34%) compared to those aged 50–69 (24%; *p* < 0.001), and OS was significantly lower (38% vs. 50%; *p* < 0.001). On the other hand, among patients with active disease, there was no significant difference in outcomes between the two age groups. Furthermore, it is important that among patients aged ≥ 70, those with a KPS of ≥80% demonstrated improved survival [126]. A recent systematic review examining transplant outcomes in patients aged ≥ 70 years with hematologic malignancies, including AML, also highlighted the importance of performance status and comorbidities (HCT-CI). In the review, the most common cause of death was disease progression, followed by infections and GVHD, suggesting that special attention should be paid to infection and GVHD risks in elderly patients [127]. Research is actively ongoing on the optimal geriatric assessment for transplant recipients and its clinical utility. Olin et al. reported that cognitive function is a predictor of NRM [128]. Furthermore, studies combining geriatric assessment with biomarkers are also underway, and their results are eagerly anticipated [129].

### 4.9. Improvements in Supportive Care and Management of Complications

In transplant for AML patients, it should not be overlooked that improvements in supportive care and the management of complications have also contributed to the expansion of transplant eligibility and better transplant outcomes. In the management of cytomegalovirus (CMV), which has traditionally relied on monitoring with CMV antigenemia or PCR and preemptive therapy, the introduction of letermovir prophylaxis has had a significant impact. Letermovir has been shown to significantly suppress CMV reactivation during the prophylactic period and contributes to a reduction in NRM [130,131]. However, reactivation following the discontinuation of prophylaxis and the high cost of the drug remain ongoing concerns. Additionally, some studies have suggested that CMV reactivation might contribute to the suppression of AML relapse [132,133,134]. There is an ongoing debate regarding which patients should receive letermovir and the optimal duration of its use.

Significant progress has been made in the management of GVHD over the past decade [135]. In addition to conventional clinical diagnosis, recent advances have enabled risk stratification using biomarkers such as ST2 and REG3α, allowing for more personalized treatment strategies based on individual patient risk [136]. Furthermore, several novel agents have become available for the treatment of steroid-refractory GVHD, including ruxolitinib (Janus kinase 1/2 inhibitor) [137,138], ibrutinib (Bruton tyrosine kinase inhibitor) [139], belumosudil (Rho-associated coiled-coil-containing protein kinase 2 inhibitor) [140], and axatilimab (humanized IgG4 monoclonal antibody) [141], which have also contributed to reducing the side effects associated with prolonged steroid use.

Sinusoidal obstruction syndrome/veno-occlusive disease (SOS/VOD) is one of the serious complications following transplant. Traditionally, the Seattle and Baltimore criteria have been used [142,143]. In recent years, the importance of early detection and intervention has been increasingly recognized. In 2016, the EBMT proposed new diagnostic criteria that were more suitable for earlier diagnosis and for identifying late-onset SOS/VOD [144], which were further revised in 2023 [145]. Additionally, the HokUS-10, a scoring system based on ultrasound, was proposed. Although it was based on data from a single center, it has drawn attention for its high sensitivity and specificity [146]. Regarding treatment, defibrotide remains the only approved agent for SOS/VOD. Early administration of defibrotide has been shown to prevent progression to multi-organ failure and improve outcomes [147]. Although the prophylactic use of defibrotide was investigated in the international, multicenter phase III HARMONY trial for high-risk patients, the study did not demonstrate a significant preventive benefit [148].

## 5. Conclusions

The evolution of transplantation over the last 20 years has been remarkable. Transplantation is no longer a last resort but has become an integral part of comprehensive treatment strategies, particularly for AML. This progress will continue, with future approaches likely to include combinations or comparisons with advanced therapies such as CAR-T or other cell therapies. Understanding the strengths and limitations of each treatment strategy will be crucial to integrating them effectively and achieving the best possible outcomes for patients.

## Figures and Tables

**Figure 1 cells-14-00572-f001:**
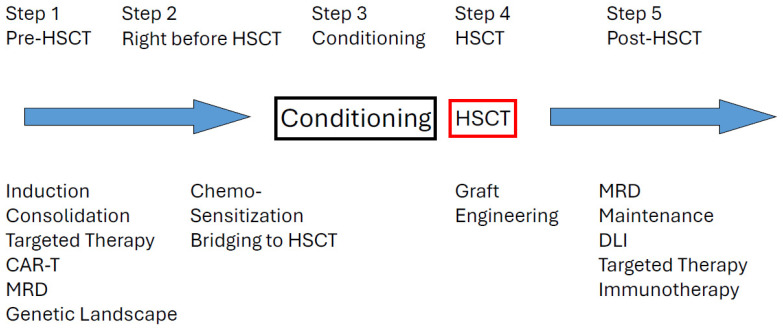
Comprehensive therapeutic strategy for hematologic malignancies. Abbreviations: CAR-T—Chimeric Antigen Receptor T-cell, DLI—Donor Lymphocyte Infusion, HSCT—Hematopoietic Stem Cell Transplantation, MRD—Minimal Residual Disease.

**Figure 2 cells-14-00572-f002:**
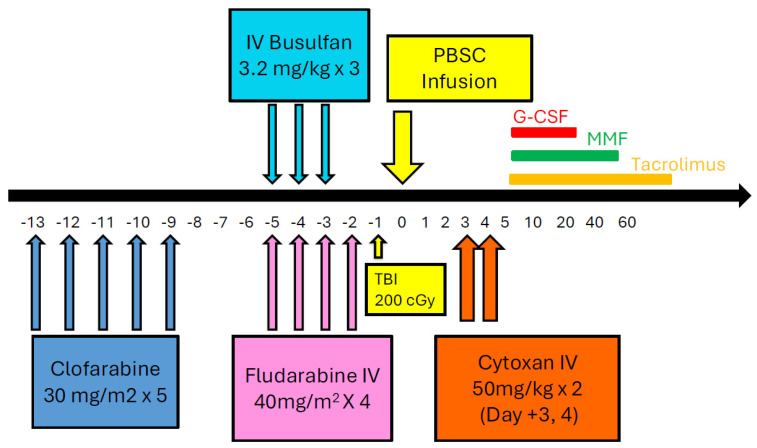
Clofarabine preconditioning regimen. FluBu3 was too toxic even for relatively young patients, so we switched to FluBu2 or FluMel with PTCy (refer to the manuscript for details). Abbreviations: Cytoxan—Cyclophosphamide, G-CSF—Granulocyte Colony-Stimulating Factor, IV—Intravenous, MMF—Mycophenolate Mofetil, PBSC—Peripheral Blood Stem Cell, TBI—Total Body Irradiation.

**Figure 3 cells-14-00572-f003:**
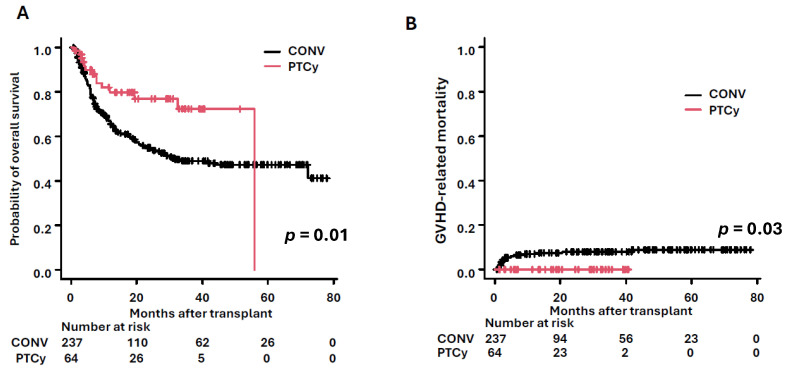
Comparison between conventional therapy and PTCy in our institution. (**A**) Overall survival and (**B**) GVHD-related mortality. The black line represents the conventional therapy group, and the red line represents the PTCy group. Abbreviations: CONV—Conventional therapy, GVHD—Graft-Versus-Host Disease, PTCy—Post-Transplant Cyclophosphamide.

**Figure 4 cells-14-00572-f004:**
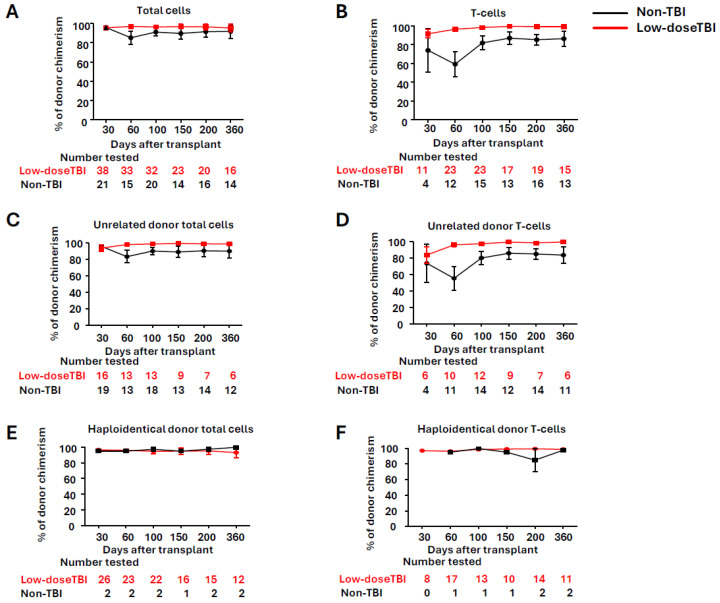
Comparison of donor chimerism kinetics between TBI and non-TBI groups in patients receiving PTCy at our institution. (**A**) Total donor chimerism in all patients, (**B**) T-cell donor chimerism in all patients, (**C**) total donor chimerism in unrelated transplant, (**D**) T-cell donor chimerism in unrelated transplant, (**E**) total donor chimerism in haploidentical transplant, and (**F**) T-cell donor chimerism in haploidentical transplant. The black line represents the non-TBI group, and the red line represents the TBI group. Abbreviations: TBI—Total Body Irradiation.

**Table 1 cells-14-00572-t001:** Methods for MRD detection in AML.

	Molecular (NGS-Based, PCR-Based)	Multi-Color Flow Cytometry	Molecular Mutation Panel (NGS-Based)	Fluorescence In Situ Hybridization	Cytogenetics
Targets	NPM1, FLT3-ITD, CBFB-MYH11, PML-RARα, etc.	Aberrant immunophenotypes	Myeloid panel (Detect around 50 leukemia-related genes)	KMT2A-MLL, Del17p, +8, −5q, etc.	NA
Strength	▪ Disease-specific ▪ Very sensitive (up to 10^−6^)	Relatively sensitive (up to 10^−3^)	Comprehensive	Easily accessible	▪ Easily accessible ▪ Comprehensive structural variant analysis
Weakness	▪ Available only for limited mutations/translocations ▪ May fail to track MRD when clonal evolution occurs	▪ Cell viability is critical ▪ Needs 500 K to 1 million CD45+ cells to achieve 0.1% sensitivity ▪ Less specific	▪ Relatively low sensitivity (up to 10^−2^) ▪ Detection of clonal hematopoiesis of indeterminate potential (CHIP) mutations may not indicate relapse	▪ Relatively low sensitivity (up to 10^−2^) ▪ Fail to detect short length of deletion or translocation	▪ Hard to identify cryptic structural variances ▪ Low sensitivity (can analyze only 20 metaphase cells)

Abbreviations: CBFB-MYH11: Core-binding factor subunit beta—Myosin heavy chain 11, FLT3-ITD: Fms-like tyrosine kinase 3—Internal tandem duplication, KMT2A-MLL: Lysine methyltransferase 2A—Mixed-lineage leukemia, MRD: Minimal residual disease, NA: Not applicable, NGS: Next-generation sequencing, NPM1: Nucleophosmin 1, PCR: Polymerase chain reaction, PML-RARα: Promyelocytic leukemia—Retinoic acid receptor alpha.

**Table 3 cells-14-00572-t003:** Method to prevent engraftment failure.

Modality	Suppress Donor T-Cells	Suppress Host T-Cells	Effect to Prevent Graft Failure	Problems	Comments
Intensifying Conditioning	No	Yes	Good	Toxicity	-
ATG	Yes	Yes	?	Infection	Works in T-cell depleted transplant
Alemtuzumab	Yes	Yes	?	Infection	Works in T-cell depleted transplant
TBI	No	Yes	Good	Toxicity	-

Abbreviations: ATG—anti-thymocyte globulin, TBI—total body irradiation, ?—unknown.

## Data Availability

No new data were created or analyzed in this study.

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
