# Peer review of "Evolution of Allogeneic Stem Cell Transplantation: Main Focus on AML"

_cells, 2025, doi:10.3390/cells14080572_

Round 1

Reviewer 1 Report

Comments and Suggestions for Authors

In the last years, we have witnessed exciting advancements in the treatment of hematologic malignancies, including acute myeloid leukemia. Consistently, allogeneic hematopoietic stem cell transplantation (allo-HSCT) has shifted from "ultimate" therapy to one of the therapeutic strategies adopted for acute myeloid leukemia.

The advancements in risk stratification, alongside conditioning regimens, and graft-versus-host disease (GVHS) prophylaxis have improved the outcome and transplant eligibility as well as donor selection.

In the manuscript titled "Evolution of allogenic stem cells transplant: main focus on AML" Inoue Y. and colleagues review the key developments in allo-HSCT for AML. 

The manuscript comprehensively covers the topic. From my side, no major flaws are detected. A minor issue concerns the references that appear not properly formatted according to the journal guidelines.

Reviewer 2 Report

Comments and Suggestions for Authors

I have the following concerns/questions:

1. TP53 AML as you state is a very challenging disease, extremely hard to treat. Are there any data regarding which subset of TP53 AML patients should be transplanted? I am giving the following reference for information, but the authors are instructed to search for more information. 

Nawas MT et al, Blood Adv 2024; 8 (3): 553-561

2. Please report that quizartinib as maintenance therapy after allo is approved in Europe (EMA), but not in the United States (FDA). 

3. Please also state that VEN-AZA or VEN combinations in general might serve as a bridge for allo transpantation in AML patients, when there is a temporal lack of a donor, or for patients with a more severe disease that can be transplanted. 

Reviewer 3 Report

Comments and Suggestions for Authors

The titel and scope seems misleading.It says evolution, but many important components contributing to better outcome in HCT in more recent years are not mentioned.The article especially focus on AML ,although other indications are also mentioned.Issues like MRD,posttransplant maintenance therapy etc are well covered.However,regarding better prevention of GVHD ,only PTCY is mentioned.Treatment and diagnostics of GVHD,SOS,infections etc are rarely mentioned but the improvments to handle these problems should be added to fulfill the expectations.The authors should add such data to fill the title.AML is a disease in patients of relative high age.The authors should discuss outcome with HCT in patients of high age.Patients above 70 years of age are now accepted for HCT in many centers.Please discuss.

Minor,Linde 308 p=0.08 is ns just a trend as mentioned regarding the same p-value in the next sentence.

in the figures add p-values!

Round 2

Reviewer 2 Report

Comments and Suggestions for Authors

I have no further concerns. 

Reviewer 3 Report

Comments and Suggestions for Authors

This article is now fine and suitable for publication.